# Bilateral Shifts in Deuterium Supply Similarly Change Physiology of the Pituitary–Thyroid Axis, but Differentially Influence Na^+^/I^−^ Symporter Production

**DOI:** 10.3390/ijms24076803

**Published:** 2023-04-06

**Authors:** Nataliya V. Yaglova, Sergey S. Obernikhin, Ekaterina P. Timokhina, Valentin V. Yaglov, Dibakhan A. Tsomartova, Svetlana V. Nazimova, Elina S. Tsomartova, Marina Y. Ivanova, Elizaveta V. Chereshneva, Tatiana A. Lomanovskaya

**Affiliations:** 1Laboratory of Endocrine System Development, A.P. Avtsyn Research Institute of Human Morphology of Federal State Budgetary Scientific Institution “Petrovsky National Research Centre of Surgery”, 119991 Moscow, Russia; ober@mail.ru (S.S.O.); rodich@mail.ru (E.P.T.);; 2Department of Human Anatomy and Histology, Federal State Funded Educational Institution, Higher Education I.M. Sechenov First Moscow State Medical University, 119435 Moscow, Russia

**Keywords:** deuterium, deuterium-depleted water, deuterium-enriched water, thyroid hormones, thyroid gland, pituitary gland

## Abstract

Deuterium, a stable isotope of hydrogen, is abundant in organisms. It is known to produce various biological effects. However, its impact in thyroid hormone synthesis and secretion is poorly studied. The aim of this investigation was to evaluate the dynamics of thyroid hormones and pituitary thyroid-stimulating hormone secretion during bilateral shifts in deuterium supply and assess a possible role of the Na^+^/I^−^ symporter (NIS), the main iodide transporter, in altered thyroid function. The experiment was performed on adult male Wistar rats, which consumed deuterium-depleted ([D] = 10 ppm) and deuterium-enriched ([D] = 500,000 ppm) water for 21 days. The assessment of total thyroxine and triiodothyronine and their free fractions, as well as thyroid-stimulating hormone in blood serum, revealed the rapid response of the thyroid gland to shifts in the deuterium/protium balance. The present investigation shows that the bilateral changes in the deuterium body content similarly modulate thyroid hormone production and functional activity of the pituitary gland, but the responses of the thyroid and pituitary glands differ. The response of the thyroid cells was to increase the synthesis of the hormones and the pituitary thyrotropes, in order to reduce the production of the thyroid-stimulating hormone. The evaluation of NIS serum levels found a gradual increase in the rats that consumed deuterium-enriched water and no differences in the group exposed to deuterium depletion. NIS levels in both groups did not correlate with thyroid hormones and pituitary thyroid-stimulating hormone production. The data obtained show that thyroid gland has a higher sensitivity to shifts in the deuterium body content than the hypothalamic–pituitary complex, which responded later but similarly in the case of deuteration or deuterium depletion. It indicates a different sensitivity of the endocrine glands to alterations in deuterium content. It suggests that thyroid hormone production rate may depend on deuterium blood/tissue and cytosol/organelle gradients, which possibly disturb the secretory process independently of the NIS.

## 1. Introduction

The role of the stable isotopes of the major biogenic elements is still poorly understood and is a promising area of biophysical, biochemical, molecular, genetic, and other research. Hydrogen has two stable isotopes, deuterium and protium. Deuterium abundance in living organisms is well documented [1,2,3]. However, the role of deuterium in metabolism is not fully understood and requires extensive research. It has been shown that shifts in the deuterium/protium balance provoke metabolic changes and affect cell proliferation and apoptosis [4,5,6,7,8]. These facts attracted a great interest from researchers and physicians, since they provide an opportunity to use the deuterium body content as an instrument in anticancer therapy. Shifts in the deuterium body content have been shown to be easily achieved by the substitution of drinking water by deuterium-depleted or deuterium-enriched water, a simple and safe method [9]. The administration of deuterium-depleted water has been shown to improve the results of antitumor treatment and has been recommended as a supplemental treatment for chemotherapy [3,10]. As the beneficial effects of deuterium-depleted water resulted from the better function of mitochondrial oxidation, deuterium reduction has potentially become a promising tool in the treatment of metabolic disorders [4,7].

Lack of knowledge on deuterium function complicates the prediction and assessment of physiological changes induced by shifts in the deuterium/protium balance. It has been found that the consumption of deuterium-depleted water from the day of inoculation of tumor cells does not affect tumor growth and metastasis in mice, whereas preliminary deuterium depletion significantly reduces tumor growth and increases survival rate [10]. These results suggest that shifts in the deuterium body content affect metabolic patterns and alter organ physiology, and that development of physiological changes depends on the time and amplitude of the deuterium/protium disbalance. Since all metabolic reactions are regulated by hormones, they may result from the impact of deuterium shifts in cell metabolic reactions as well as altered endocrine gland physiology. Changes in the deuterium body content, in this case, may be used for the modulation of endocrine gland function, for example, in the treatment of severe somatic disorders such as systemic inflammatory response and sepsis.

Despite numerous studies on thyroid hormone synthesis and secretion, its regulation by endogenous and exogenous factors is still to be studied. The least investigated field of thyroid research is the impact of isotopes of biogenic atoms on thyroid hormone secretion and function. The thyroid gland is an evolutionarily ancient organ that has adapted to many changes in the environment. At the same time, hormone synthesis in the thyroid gland is the most sensitive to the ion-transporting channels, primarily iodine transporters, and the intensity of redox processes, which are influenced by the deuterium content of the cells.

Deuterium is known to form stronger and more resistant cleavage bonds with other atoms than protium [11]. It suggests that deuterium content may regulate the intensity of catabolic reactions and endocrine control of catabolism mediated by thyroid hormones. Deuterium/protium disbalance may also directly affect the thyroid gland. Nowadays, thyroid disorders are recognized as the most common endocrine diseases worldwide [12,13]. Thyroid dysfunction has multifactorial etiology and results from iodine deficit, exposure to negative occupational factors, and genetic predisposition. The thyroid is very sensitive to environmental and anthropogenic hazards, especially to endocrine-disrupting chemicals and ionizing radiation [14,15,16,17,18]. Therefore, the elucidation of the role of deuterium content in the function of the hypothalamic–pituitary–thyroid axis could also reveal some mechanisms of preventing or correcting thyroid dysfunction. Thyroid function is known to depend primarily on iodine supply [18]. Iodine uptake is mediated by the Na^+^/I^−^ symporter (NIS), a protein located in the basolateral membranes of thyroid follicular cells. The iodine transport mechanisms have been shown to be closely submitted to the regulation of NIS expression. The aim of the present study was to evaluate the dynamics of thyroid and pituitary hormone secretion and assess a possible role of NIS, the main iodide transporter, in altered thyroid function induced by bilateral shifts in the deuterium supply.

## 2. Results

### 2.1. Body Weight Parameters

Since young postpubertal rats were enrolled in the investigation, physiological gain in body weight was observed at the end of the experiment. The rats in the control group demonstrated a 38% increase in body weight by the 21st day. The rats that consumed deuterium-depleted water increased their weight by 33%. The rats that consumed deuterium-enriched water showed the smallest increase in body weight—9.7%, and their weight on the 21st day of the experiment was significantly lower than in the control group and the group that consumed deuterium-depleted water (Figure 1).

### 2.2. Water Consumption Parameters

The average daily consumption of rats that consumed deuterium-depleted water per kg of body weight did not differ from the values of the control group. The consumption of deuterium-enriched water by the rats was slightly lower than in the control, but the differences were statistically insignificant (Figure 2).

### 2.3. Thyroid Hormone Profiles

#### 2.3.1. Before the Exposure

The total concentrations ofT3 and T4, their free fractions, and thyroid-stimulating hormone (TSH) were measured the day before the exposure (day 0). No significant differences between the groups were found (Figure 3 and Figure 4).

#### 2.3.2. First Day of Exposure

The thyroid hormone profile of the rats that consumed deuterium-depleted water differed from the control values. The serum concentration of T4 was elevated and the fT4 level was decreased (Figure 3A,B). T3 and fT3 serum content were insignificantly diminished (Figure 3C,D). The TSH level was significantly lower than in the control (Figure 4).

In rats that consumed deuterium-enriched water, the T4 level was also elevated, while the fT4 concentration was in the control range (Figure 3A,B). Changes in T3 concentration were similar, and the decrease in its free fraction was more pronounced (Figure 3C,D). The TSH was also decreased compared to the control, and did not differ from the previous group (Figure 4).

#### 2.3.3. Third Day of Exposure

In the rats that consumed deuterium-depleted water, the content of T4 and T3 in the serum did not differ from the control, but the concentrations of their free fractions increased significantly and exceeded the control values. The TSH level increased significantly, but did not exceed the values of the control group (Figure 3 and Figure 4).

The changes in the thyroid status of rats that consumed deuterium-enriched water were similar, but more pronounced. There was an increase in the production of T4 and T3, as well as their free fractions. The concentrations of these hormones significantly exceeded the values of the control group. Compared with the group that consumed deuterium-depleted water, there was a higher content of T4, fT4, and T3. The TSH level was also increased, but to a much lesser extent, and it was lower than in the control group and the group that consumed deuterium-depleted water (Figure 3 and Figure 4).

#### 2.3.4. Seventh Day of Exposure

The rats that consumed deuterium-depleted water had the same levels of thyroid hormones as the control animals. However, their TSH concentration was half that of the control values (Figure 3 and Figure 4).

In rats that consumed deuterium-enriched water, the concentration of T4 and fT4 did not differ from the control values. Unlike the previous group, a significant decrease in T3 concentration was revealed. At the same time, the level of fT3 did not change compared to the third day and did not differ from the control values. The TSH level was also lower than the control values, but higher than in rats that consumed deuterium-depleted water (Figure 3 and Figure 4).

#### 2.3.5. 14th Day of Exposure

On the 14th day of the experiment, the rats that consumed deuterium-depleted water showed a significant decrease in the concentration of all thyroid hormones. The reduction in total T4 and T3 levels was more pronounced than that in their free fractions. The TSH serum content was three times lower than the control values. Similar changes in thyroid hormone profiles were found in rats that consumed deuterium-enriched water (Figure 3 and Figure 4).

#### 2.3.6. 21st Day of Exposure

On the 21st day of the experiment, the restoration of thyroid hormone secretion was noted in rats that consumed deuterium-depleted water. T4 and fT4 concentrations, as well as T3, corresponded to the control values, and fT3 even exceeded them. The TSH level was lower than that in the control (Figure 3 and Figure 4).

A similar trend was observed in the rats that consumed deuterium-enriched water. The T4 concentration corresponded to the control values, and the T3 level was insignificantly lower than in the control, but significantly lower than in rats that consumed deuterium-depleted water. The concentration of fT4 was the lowest among the compared groups. The fT3 levels exceeded the control values and did not differ from the values of the group that consumed deuterium-depleted water. The level of TSH was also significantly reduced (Figure 3 and Figure 4).

### 2.4. Serum Levels of Sodium/Iodide Symporter

#### 2.4.1. Before the Exposure

Before the exposure, the serum levels of NIS did not differ between the compared groups (Figure 5).

#### 2.4.2. First Day of Exposure

The evaluation of serum NIS levels showed no differences between the control rats and the rats that consumed deuterium-depleted water and a significant raise in NIS in the rats that consumed deuterium-enriched water (Figure 5).

#### 2.4.3. Third Day of Exposure

The rats that consumed deuterium-depleted water had similar NIS levels to the control group, unlike the rats that consumed deuterium-depleted water. The latter had a significantly higher concentration of NIS than the control group (Figure 5).

#### 2.4.4. Seventh Day of Exposure

As in the previous study period, the NIS values of the control group and the group consuming deuterium-depleted water did not differ. In the group that consumed deuterium-enriched water, the NIS concentration was higher than in the control and the comparison group (Figure 5).

#### 2.4.5. 14th Day of Exposure

The rats that consumed deuterium-depleted water showed an insignificant reduction in NYS serum concentration. In contrast, the rats consuming deuterium-enriched water showed increased NIS production (Figure 5).

#### 2.4.6. 21st Day of Exposure

No significant changes in NIS serum levels were found in the rats that consumed deuterium-depleted water, while the group of deuterium-enriched water showed a pronounced increase in NIS levels (Figure 5). The total increase in NIS concentration from day 0 to day 21 was 60%.

## 3. Discussion

The examination of physiological parameters did not reveal significant differences in the daily intake of water with modified deuterium content. The assessment of body weight gain and water intake showed that the consumption of deuterium-depleted water was similar in volume to the control group and did not affect the physiological growth of animals. In rats that consumed deuterium-enriched water, on the contrary, the increase in body weight was insignificant. These facts are consistent with the reported data on the negative effects of high D_2_O concentrations on cell metabolism and proliferation [19,20,21,22]. Thus, in rats that consumed deuterium-enriched water, there was a factor that could affect the normal physiology of the endocrine system or be caused by endocrine, including thyroid, dysfunction.

Changes in thyroid hormone profiles found after the first day of the experiment indicate that the thyroid gland is sensitive to shifts in deuterium content. It is very important to note that the thyroid gland responded to changes in deuterium concentration mainly in blood, lympha, and, concomitantly, in the cytosol of thyroid cells, which have been shown to develop significantly faster than in solid tissues since the elimination of deuterium or deuteration in biopolymers requires much time [9,23]. This means that the intensity in thyroid hormone secretion is associated with an altered deuterium blood/tissue gradient. Early changes in thyroid hormone levels were similar in both groups and corresponded with the previously reported activation of thyroid hormone production after 24 h of consumption of both deuterium-enriched and -depleted water [24]. The return to a normal level of thyroid hormone production was mediated by the adequate response of the pituitary gland to an excess of thyroid hormones. This indicates that the negative feedback between the TSH and thyroid hormone synthesis was not affected by shifts in the deuterium supply. On the seventh day of the experiment, we found initial signs of pituitary dysfunction. The inappropriate downregulation of TSH production was revealed both after deuteration and deuterium depletion. This suggests an affection of hypothalamic or pituitary cells. On the 14th day, we observed secondary hypothyroidism typical for thyroid profiles in rats that consumed deuterium-depleted as well as deuterium-enriched water. The only difference found was in the time-period of pituitary dysfunction development. The rats that consumed deuterium-depleted water showed a slower rate of downregulation of the pituitary–thyroid axis. The affection of the hypothalamic–pituitary complex was transient, and after the 21st day, we observed changes indicative of the restoration of pituitary activity and negative feedback to thyroid hormones in both experimental groups. The rates of restoration were also found to differ. The rats that consumed deuterium-depleted water had restored normal pituitary–thyroid axis function faster than those that consumed deuterium-enriched water. Thus, in rats that consumed deuterium-depleted water, central hypothyroidism developed slower and the restoration of thyroid hormone secretion was faster than in rats that consumed deuterium-enriched water. The lower rate of restoration of thyroid function in the latter group could probably result from the growing toxicity of 50% D_2_O. In general, the changes in the rats that consumed deuterium-enriched water at all time periods were similar to those that consumed deuterium-depleted water. Thus, opposite shifts in the deuterium body content did not cause opposite changes in thyroid hormone secretion. This means that the hypothalamic–pituitary–thyroid axis is sensitive to shifts in the balance of deuterium and protium, but the direction of the shift does not matter.

It is known that small bilateral changes in deuterium content show opposite effects on cell proliferation. Deuterium depletion contributes to the inhibition of mitosis, and deuteration stimulates cell division, while an increase in deuterium content above 25% inhibits mitosis [5,25,26]. It is possible that the similarity in changes in the production of thyroid hormones also has the same dose-dependent pattern. However, shifts in deuterium content in tissues take time, and the effects observed in the first days, when changes occur in the blood, indicate the same mechanism of regulation of thyroid hormonogenesis. It has been shown that a decrease in deuterium content causes redox disbalance and oxidative stress, and an increase has an antioxidant effect [5,25]. Thyroid cells have been shown to have a high activity of redox reactions, since the key step in hormone synthesis—iodination of the thyroglobulin molecule is provided by the hydrogen-peroxide-generating system [27,28]. Therefore, it can be assumed that non-redox reactions determine the similarity in changes in hormone production during the ongoing shifts in deuterium body content.

Deuterium has been shown to produce a kinetic isotope effect, which consists in the formation of stronger chemical bonds with other atoms, and, accordingly, in altered kinetics of chemical reactions, especially of cleavage and redox reactions [29,30]. Both types of reactions are extremely important for the production of thyroid hormones, since the iodination of thyroglobulin is provided by the redox process and the formation of T4 and T3 by the thyroglobulin proteolysis reactions. A 50% D_2_O solution is a powerful source of deuterium, and its consumption for 3 weeks provides significant deuteration of the body. However, we did not reveal a decrease in the serum concentration of iodinated thyronines by the 21st day, despite the fact that the negative consequence of deuteration—a slowdown in body weight gain—manifests itself clearly. The present findings suggest that shifts in deuterium water content generate a hydrogen isotope gradient between cells and the internal environment, and then between cytosol and organelles, which affects secretory process in thyroid cells.

Thyroid hormone secretion is known to have both circadian and infradian rhythms [31,32,33]. We also observed various thyroid hormone levels in the control rats during the investigation. The free fractions of T4 and T3 demonstrated similar fluctuations. Free T4 and T3 represent fractions that are available for penetration into cells, and their concentration mainly depends on the amount of hormone-transporting proteins [34,35]. The quantification of the free T4 and T3 in the exposed rats showed that their concentrations did not undergo significant changes, in contrast to the control group; that is, the level of the unbound hormones became more stable both with a decrease and an increase in the deuterium body content. It indicates that shifts in the deuterium/protium balance do not significantly affect either the production of plasma-binding proteins or their binding capacity. Less fluctuations in free T4 and T3 suggest changes in the parameters of infradian rhythms of thyroid hormone secretion and require additional study.

It is well known that thyroid function depends on iodine supply. The transportation of iodide to thyroid cells is mediated by NIS, a protein located on the basolateral membranes of thyroid cells as well as cells of salivary glands, gastric mucosa, and renal cells [36,37]. Some chemicals such as perchlorates or dichlorodiphenyltrichloroethane have been shown to inhibit NIS and thyroid function [14,38]. The reduction in blood NIS levels is usually associated with the diminished production of thyroid hormones, and the restoration of NIS expression positively influences thyroid gland activity [39]. That is why we assessed NIS serum levels in the present investigation and surprisingly found no association with changes in thyroid hormone concentrations. Moreover, NIS did not correlate with TSH, the main positive regulator of NIS expression [40]. We observed clear differences in NIS levels between the experimental groups. The rats that consumed deuterium-depleted water showed no differences from the control values. The prolonged consumption of deuterium-enriched water evoked a gradual increase in NIS serum levels. This may have been associated with changes in the transporting activity of the iodide channel. NIS is known to transport iodide using an electrochemical gradient generated by Na,K-ATPase [36,37,38]. The deuteration of cells was previously shown to slow down the rate of ion transporters, inhibit the activity of ATP synthase, and reduce ATP production [8,41]. The elevation of NIS levels in the rats consuming deuterium-enriched water, therefore, may be considered a compensatory process and may reflect changes induced by the blood/tissue gradient and energy-associated decreasing efficacy of NIS. However, this hypothesis indicates the presence of a new unknown mechanism for the regulation of NIS synthesis, which requires further research.

## 4. Materials and Methods

### 4.1. Animals and Experimental Design

Male Wistar rats aged 6 months, weighing 350–360 g (*n* = 30), were purchased from Scientific Center of Biomedical Technologies of Federal Medical and Biological Agency of Russia (Pushchino, Moscow region, Russia). The rats were housed (3–4 rats in cage) at +22–23 °C with a 12/12 h light–dark cycle and provided with a pelleted standard chow and water *ad libitum*. The rats were enrolled in the experiment after a two-week adaptation to the local vivarium.

The rats were randomized into three groups. The control group (*n* = 10) consumed distilled tap water with a normal deuterium content (146 ppm). The 1st experimental group (*n* = 10) consumed deuterium-depleted water with [D] = 10 ppm, and the 2nd experimental group (*n* = 10) consumed deuterium-enriched water with [D] = 500,000 ppm ad libitum instead of tap water for 21 days. Venous blood sampling and weight measurement were performed under zoletil anesthesia at 10–11 a.m. the day before the experiment and on the 1st, 3d, 7th, 14th, and 21st days. After incubation at room temperature, blood cloths were separated and serum was collected. The total change in body weight was calculated on the 21st day. The volume of the consumed water was measured daily for terms 1 and 3 days, then every 3 days for terms 7, 14, and 21 days. The amount of consumed water per 1kg body weight was calculated.

### 4.2. Reagents

Deuterium-enriched water with 500,000 ppm and deuterium-depleted water with 10 ppm deuterium concentration were manufactured by B. P. Konstantinov St. Petersburg Nuclear Physics Institute of National Research Center “Kurchatov Institute” (St. Petersburg, Russia). The concentration of deuterium in the water samples was verified by a T-LWIA-45-EP isotope analyzer (Los Gatos Research Inc., San Jose, CA, USA), which determines deuterium content with an accuracy of 1 ppm.

### 4.3. Ethical Approval

The investigation was performed in accordance with the handling standards and rules of laboratory animals as consistent with “International Guidelines for Biomedical Researches with Animals” (1985), laboratory routine standards in the Russian Federation (Order of Ministry of Healthcare of the Russian Federation dated 19 June 2003 No. 267), the “Animal Cruelty Protection Act” dated 1 December 1999, and the regulations on experimental animal operation approved by the Order of Ministry of Healthcare of USSR No.577 dated 12 August 1977. Animal experiments were approved by the Ethics committee of the Research Institute of Human Morphology (protocol N 23, 25 March 2020).

### 4.4. Thyroid Profile Assessment

Total thyroxine (T4), free thyroxine (fT4), total triiodothyronine (T3), free triiodothyronine (fT3) (Cusabio, Wuhan, China), thyroid-stimulating hormone (BioVendor, Brno, Czech Republic), and NIS (Fine test, Wuchan, China) serum concentrations were measured by an enzyme-linked immunosorbent assay according to the manufacturer’s protocols with an “Anthos 2010” microplate reader at 450 nm. All samples were measured in duplicate.

### 4.5. Statistical Analysis

Statistical analyses were carried out using the software package “Statistica 7.0” (StatSoft, Tulsa, OK, USA). Normality of distribution was conferred by Shapiro–Wilk test. The central tendency and dispersion of quantitative traits with an approximately normal distribution were presented as the mean and standard deviation (M ± S.E.M.). Quantitative comparisons of independent groups were performed using ANOVA and the Duncan test for post hoc comparison. Differences were considered statistically significant at *p* < 0.05.

## 5. Conclusions

This is the first report on the dynamics of thyroid hormone secretion and pituitary control of thyroid function during the consumption of deuterium-depleted and deuterium-enriched water. The present investigation shows that bilateral changes in deuterium body content similarly modulate thyroid hormone production and the functional activity of the pituitary gland. The thyroid gland demonstrates a higher sensitivity to shifts in hydrogen isotope balance than the hypothalamic–pituitary complex, which responds to shifts in the deuterium content later than the thyroid. Unlike pituitary thyrotropes, which reduced hormone production in response to changes in deuterium content, thyroid cells responded by increasing hormone secretion. This suggests that thyroid cell physiology and hormone production rate may depend on deuterium blood/tissue and cytosol/organelles gradients. The depletion of deuterium body content had no effect on Na^+^/I^−^ symporter levels, while deuteration increased it production. The present findings do not consider the Na^+^/I^−^ symporter a key mechanism for changes in thyroid activity. The results of the study demonstrate the interdependence of thyroid hormone production and deuterium content, opening a new field in the investigation of stable isotope impact in the physiology of endocrine glands.

## Figures and Tables

**Figure 1 ijms-24-06803-f001:**
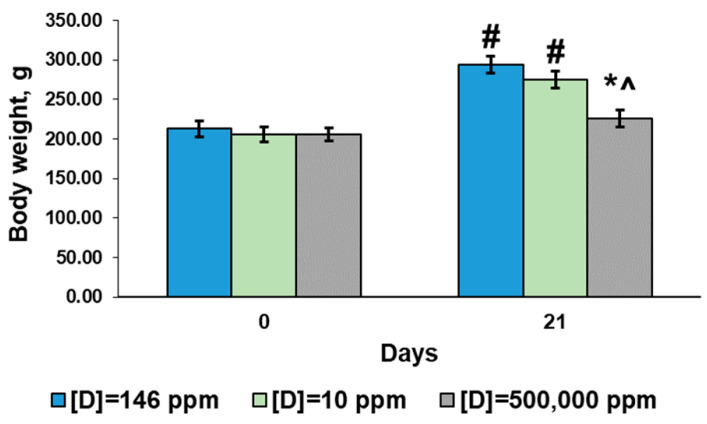
Body weight of the rats that consumed water with modified deuterium content for 21 days. Data are shown as mean ± S.E.M.; *p* < 0.05 compared to the control (*), to the rats that consumed water with [D] = 10 ppm (^), and to the initial weight (#).

**Figure 2 ijms-24-06803-f002:**
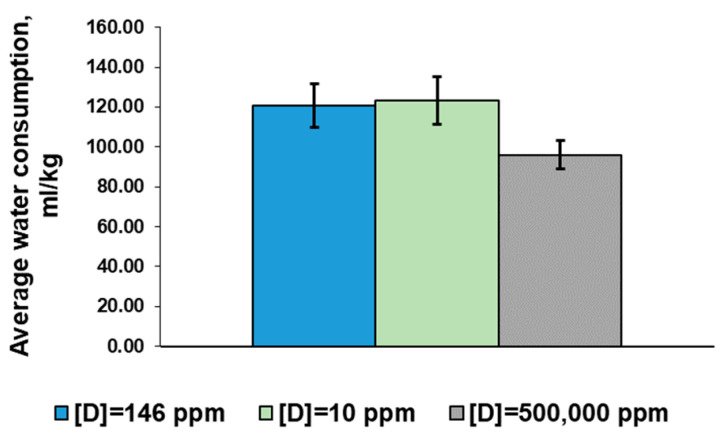
Average daily consumption of water with modified deuterium content by the rats per kg body weight. Data are shown as mean ± S.E.M.

**Figure 3 ijms-24-06803-f003:**
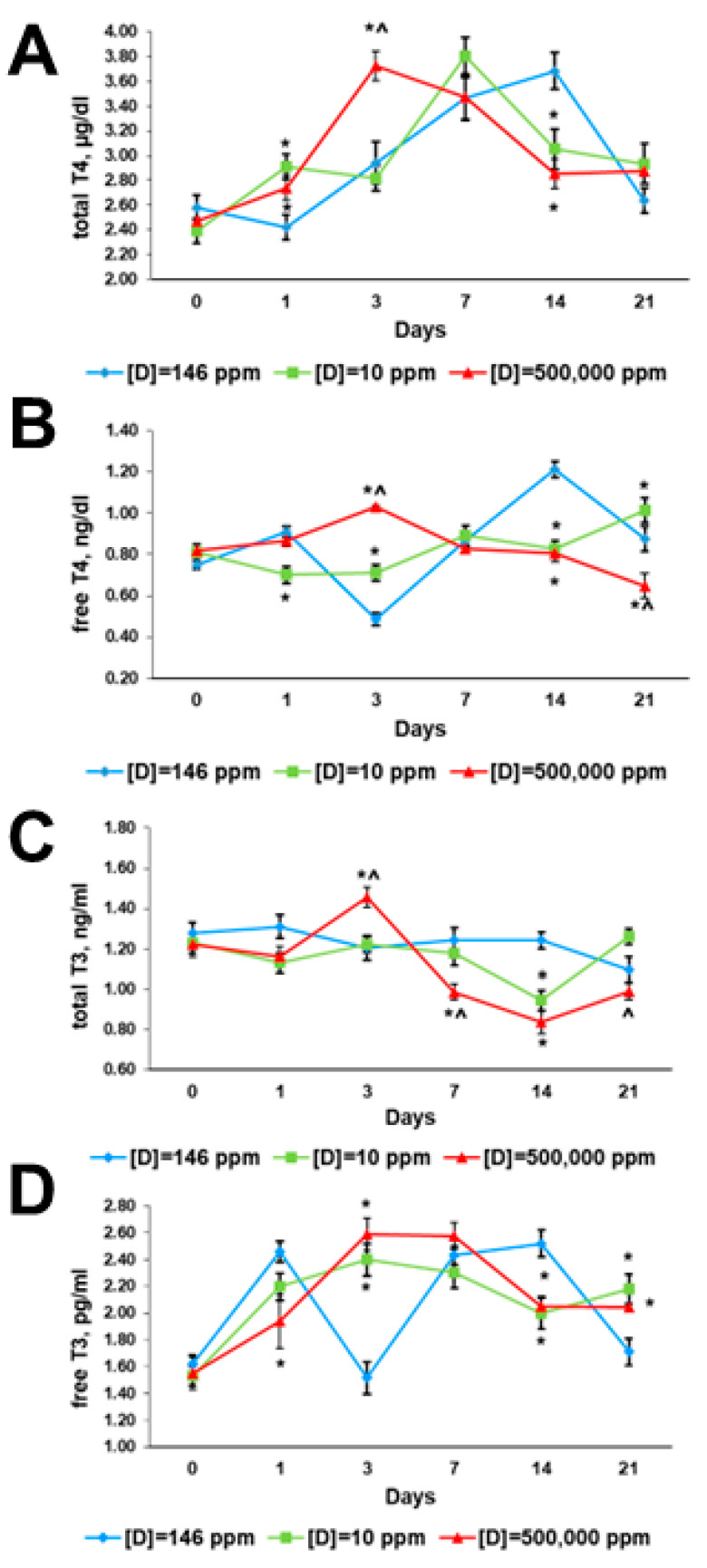
Thyroid hormone profiles of the rats that consumed water with modified deuterium content for 21 days. (**A**) Total T4 serum concentrations; (**B**) free T4 serum concentrations; (**C**) total T3 serum concentrations; (**D**) free T3 serum concentrations. Data are shown as mean ± S.E.M.; *p* < 0.05 compared to the control (*) and to the rats that consumed water with [D] = 10 ppm (^).

**Figure 4 ijms-24-06803-f004:**
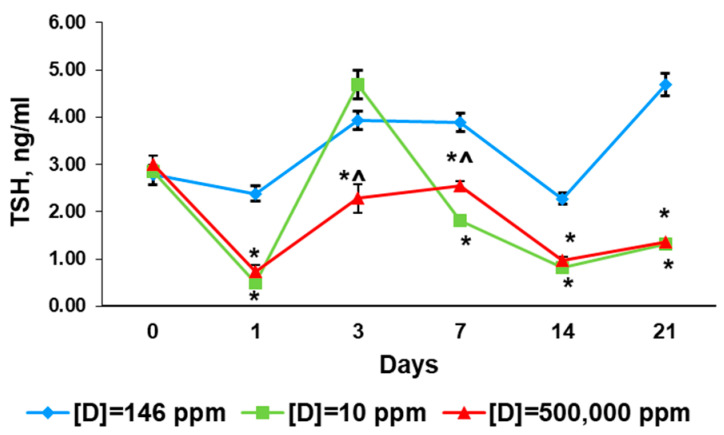
Thyroid-stimulating hormone concentrations in the rats that consumed water with modified deuterium content for 21 days. Data are shown as mean ± S.E.M.; *p* < 0.05 compared to the control (*), and to the rats that consumed water with [D] = 10 ppm (^).

**Figure 5 ijms-24-06803-f005:**
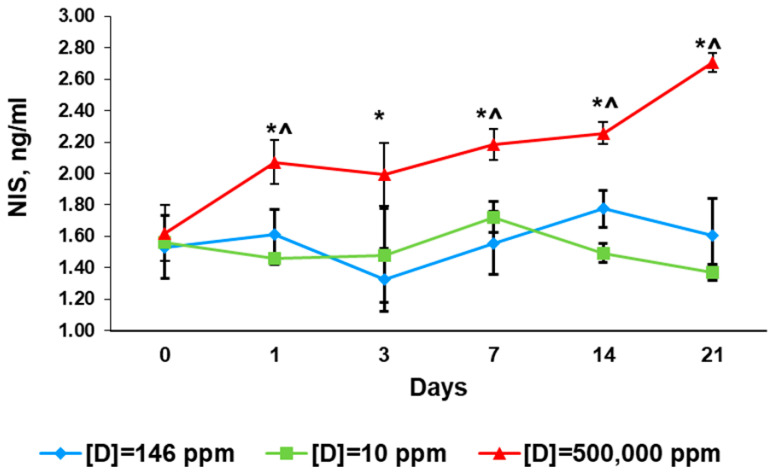
Serum concentration of NIS in the rats that consumed water with modified deuterium content for 21 days. Data are shown as mean ± S.E.M.; *p* < 0.05 compared to the control (*) and to the rats that consumed water with [D] = 10 ppm (^).

## Data Availability

The data presented in this study are available from the corresponding author upon reasonable request.

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
