# Peer review of "Bilateral Shifts in Deuterium Supply Similarly Change Physiology of the Pituitary–Thyroid Axis, but Differentially Influence Na+/I− Symporter Production"

_ijms, 2023, doi:10.3390/ijms24076803_

Round 1
Reviewer 1 Report
The manuscript submitted by Yaglova et al., is an interesting in vivo study investigating the Bilateral shifts in deuterium supply and how they affect physiology of the pituitary-thyroid axis and influence Na+/I- symporter production. The study is informative and is dealing with a fundamental question as per the effect of deuterium on thyroid function.
The reviewer would like to raise the following points for authors to consider:
1. The introduction alludes to the importance and potential clinical significance of the work herein but it is not explicit. It would be helpful to illustrate how the generated knowledge of this work could inform, predictive, diagnostic and clinical practices in general for promoting health in humans.
2. How was the number of animals determined (power calculation etc)?
3. The authors state that they housed 3-4 rats in a cage. Was there consistency in the holding conditions? There may be confounders if there are varied numbers of animals held per cage.
4. Proofreading for grammar, syntax improvement and typos is highly recommended.
Good job overall.
Reviewer 2 Report
The manuscript is correct, the experimental setup is good, although really simple. It is written in a very simple way, without drawing any conclusions until the Discussion. I would suggest to draw conclusions in the Results section since it is a pure descriptive article.
Author Response
Dear Reviewer! We thank you for your work in reviewing our manuscript and the positive assessment of our investigation.
Comment 1: The manuscript is correct, the experimental setup is good, although really simple. It is written in a very simple way, without drawing any conclusions until the Discussion. I would suggest to draw conclusions in the Results section since it is a pure descriptive article.
Response 1: We have written the article strictly according to the Recommendations for Authors of the journal and put the conclusion after discussion.